# Elevated Kappa Index in the Absence of Cerebrospinal Fluid IgG Oligoclonal Bands: Contribution of Intrathecal IgM and IgA Synthesis

**DOI:** 10.3390/biom15010090

**Published:** 2025-01-09

**Authors:** Krzysztof Smolik, Roberta Bedin, Patrizia Natali, Martina Cardi, Diego Franciotta, Anna Maria Simone, Paolo Immovilli, Mario Santangelo, Matteo Gastaldi, Giulia De Napoli, Francesca Vitetta, Diana Ferraro

**Affiliations:** 1Department of Biomedical, Metabolic and Neurosciences, University of Modena and Reggio Emilia, 41121 Modena, Italy; 198506@studenti.unimore.it (K.S.); roberta.bedin@unimore.it (R.B.);; 2Department of Laboratory Medicine, Ospedale Civile di Baggiovara, Azienda Ospedaliero-Universitaria and Azienda Unità Sanitaria Locale, 41124 Modena, Italy; p.natali@ausl.mo.it; 3Department of Clinical Pathology, Santa Chiara Hospital, Azienda Provinciale per I Servizi Sanitari (APSS), 38122 Trento, Italy; diego.franciotta@mondino.it; 4Neurology Unit, Ramazzini Hospital, 41012 Carpi, Italym.santangelo@ausl.mo.it (M.S.); 5Neurology Unit, G. da Saliceto Hospital, 29121 Piacenza, Italy; paolo.immovilli.md@gmail.com; 6Neuroimmunology Laboratory, IRCCS Mondino Foundation, 27100 Pavia, Italy; matteo.gastaldi@mondino.it; 7Neurology Unit, Ospedale Civile Baggiovara, Azienda Ospedaliero Universitaria di Modena, 41125 Modena, Italy; vitetta.francesca@aou.mo.it

**Keywords:** kappa index, oligoclonal bands, IgM/IgA intrathecal synthesis

## Abstract

The kappa index is a well-established marker of intrathecal synthesis (IS) of immunoglobulin (Ig). Routinely used for diagnostic aims, IgG IS, which can be assessed quantitatively (ad hoc formulas) or qualitatively (oligoclonal bands, OCBs), may fail in detecting a humoral immune response within the central nervous system (CNS). The main aim of this study was to evaluate the kappa index for its ability to detect the presence of CNS humoral immunity and to associate it with a distinct group of disorders, in the absence of IgG IS/OCBs. Within the kappa index-positive, IgG OCB-negative (Kappa+OCB-) patient group, we also examined whether IgM/IgA IS, determined with the IgM/IgA index and CSF IgM OCBs, could contribute to disease group stratification. Diagnoses were classified as multiple sclerosis (MS), or other inflammatory (INFL), infectious (INFECT), or non-inflammatory (Other) central/peripheral nervous system disorders. Sixty-nine Kappa+OCB- patients and 50 controls (24 Kappa-OCB- and 26 Kappa+OCB+ patients) were included in this study. The most frequent diagnosis in the Kappa+OCB- group was MS (27/69), followed by INFECT (16/69). Additional evidence of IS was demonstrated through an elevated IgG/IgM/IgA index or by the presence of IgM OCBs in 59%, and through only IgM/IgA IS in 52% of cases. In INFECT patients, the median IgM/IgA indexes were higher (*p* < 0.001) than in other groups, with 18 patients (95%) presenting an elevated IgM index, 11 patients (58%) presenting CSF IgM OCBs, and 10 patients (53%) presenting an elevated IgA index. The vast majority of all INFECT (16/19) belonged to the Kappa+OCB- group. Our data confirm that the kappa index performs at the highest level in assessing intrathecal humoral immunity and supporting the diagnosis of both MS and CNS infectious disorders, which are also characterized by the intrathecal production of IgM and IgA.

## 1. Introduction

The kappa index (cerebrospinal fluid (CSF)/serum kappa free light chains (KFLCs) ratio divided by the CSF/serum albumin ratio) has been gaining increasing interest as a marker of the intrathecal synthesis (IS) of immunoglobulins and, in particular, in the diagnostic work-up of multiple sclerosis (MS). According to most studies, it is more sensitive, though slightly less specific compared with IgG oligoclonal bands (OCBs) for an MS diagnosis, with a similar diagnostic accuracy [1,2,3]. Other advantages of the kappa index are its quantitative, automated/non-operator-dependent output, its rapidity and lower cost compared with OCB detection, and its good agreement across different platforms and assays [4]. Consequently, it has been proposed to include the kappa index as a valid alternative to OCBs in the upcoming revision of McDonald’s diagnostic criteria for MS [3]. Contrarily to OCBs, which only capture an IS of IgG, an elevated kappa index may reflect an IS of Ig classes other than IgG, since KFLCs also constitute IgM, IgA, IgD, and IgE, and the kappa index can thus be elevated in the absence of intrathecal IgG production but in the presence of central nervous system (CNS)-colonizing Ig-producing cells.

IgM IS has been mostly described in MS patients, where it has been linked to a worse prognosis [5,6,7,8], and in CNS infections. In particular, the IS of IgM may represent a primary response in infectious CNS disorders [9]. As shown in a large study on 4026 paired serum/CSF samples of patients from the outpatient clinic of the University Hospital Basel, including 293 MS patients, a dominant IgM IS was found in neuroborreliosis, mumps–meningoencephalitis, and non-Hodgkin’s lymphomas with CNS involvement. IgA synthesis, on the other hand, prevailed in neurotuberculosis, brain abscess, and adrenoleukodystrophy. Overall, an isolated IS of IgM or IgA was found in 33 and 14% of cases, respectively [10].

Since the use of the kappa index has been introduced in routine clinical practice only in recent years, and its use is, to date, not widespread, the aim of the present study was to collect information on patients with an elevated kappa index in the absence of IgG OCBs to aid the clinician in the interpretation of this result. In particular, we ascertained the diagnoses associated with this finding and assessed the intrathecal synthesis of IgM and IgA in these patients using linear and hyperbolic indexes of IS. Furthermore, the data were compared with those of two control groups: patients with an elevated kappa index and IgG OCBs (Kappa+OCB+) and patients without an elevated kappa index and absent CSF IgG OCBs (Kappa-OCB-). As a secondary outcome, we compared laboratory data between the different diagnostic categories, irrespective of their kappa index/OCB status.

## 2. Materials and Methods

### 2.1. Population

Of all consecutive patients undergoing a diagnostic spinal tap for any reason between September 2018 and September 2020, and whose CSF and serum were analysed for the presence of IgG OCBs using IEF, and for the determination of kappa free light chains (using the Optilite turbidimetric analyser, The Binding Site Group, Birmingham, UK), we selected those patients with a kappa index of ≥5.00 and absent CSF IgG OCBs. The cut-off of 5.00 was chosen as an intermediate value between the best cut-off determined in a recent meta-analysis for a MS diagnosis (6.10) [1] and the best cut-off for the prediction of a CNS inflammatory disease according to a previous study in our laboratory (3.90) [2].

As controls, we randomly selected of a group of patients with a kappa index of ≥5.00 and CSF IgG OCBs, and a group of patients with a kappa index of <5.00 and absent CSF IgG OCBs.

All patients’ charts were reviewed to ascertain their final diagnosis.

The study was approved by the Modena Ethics Committee (protocol No. 1113/2019; date of approval: 14 January 2020).

### 2.2. Laboratory Analyses

Analyses were carried out on samples stored in cryogenic tubes at −80 °C after sampling. Serum IgM, CSF IgM, serum albumin, and CSF albumin were analysed on the Optilite turbidimetric analyser (The Binding Site Group, Birmingham, UK). Stored samples were also analysed for the presence of IgM OCBs by means of agarose gel IEF, followed by immunoblotting with polyclonal specific anti-human IgM antibodies (Dako, Santa Clara, CA, USA), a qualitative method developed in-house, devised by modifying the procedure proposed by Villar et al. [11]. A description of the modifications applied to the original procedure can be found in the Appendix A.

We calculated the linear IgG, IgM, and IgA indexes, and patients were considered to have an IS of IgG/IgM/IgA in the case of values above 0.70 for IgG, 0.10 for IgM, and 0.40 for IgA. Furthermore, we applicated Reiber’s formulas for each of the three classes of patients, according to their OCB and kappa index status (Kappa+OCB-, Kappa-OCB-, and Kappa+OCB+), to take the non-linear relationship of the blood-to-CSF transfer between albumin and immunoglobulins into account [12]. The Reibergrams were created using the free software available at Albaum (www.albaum.it, accessed on 18 June 2024).

### 2.3. Statistical Methods

Comparisons among groups were made using the chi-square test or Fisher’s exact test, as appropriate, and the Kruskal–Wallis test with post hoc comparisons. The diagnostic accuracy of the IgM/IgA index was assessed using sensitivity, specificity, negative/positive predictive value (NPV/PPV), receiver operating characteristics (ROC) curves, and the Youden index to find the best cut-off. Statistical analyses were performed using STATA 16 (StataCorp, College Station, TX, USA).

## 3. Results

### 3.1. Study Population and Associated Diagnoses

We enrolled a total of 119 patients: 69 were Kappa+OCB-, 24 were Kappa-OCB-, and 26 were Kappa+OCB+. Diagnoses associated with a Kappa+OCB- status are shown in Table 1, while the frequency of diagnoses in the overall population and in the Kappa-OCB- and Kappa+OCB+ groups are shown in Appendix A.

### 3.2. Demographic and Laboratory Findings Across Patient Groups

The demographic and laboratory findings of the overall population and of patients grouped based on their kappa index/OCB status are shown in Table 2.

A total of 52% of the Kappa+OCB- patients showed signs of IS through a positive IgM/IgA index or by IgM OCB. This percentage increased to 59% when including patients with a positive IgG index.

As regards IF in Kappa+OCB- patients, 14 of them (20%) showed IS of IgM/IgA (IF > 0), with the percentage increasing to 33% when considering IgG IS (Table 3).

Reibergrams showing the patients’ CSF/serum IgG/M/A quotients (Q Ig) can be found in Figure 1. As can be seen, in the Kappa-OCB- group, most values very rarely fall above the upper limit. In the Kappa+OCB+ group (mostly MS patients), there is the largest proportion of patients with IS of IgG, while IS of IgM and IgA is mostly seen in the Kappa+OCB- group.

The linear IgG index and hyperbolic Q_IgG_ showed a very similar performance, although there were some discordant cases: the IgG index was positive in five patients with absent IgG IF (two INFL and three MS), all of whom had an elevated kappa index. Contrarily, four patients (three MS and one INFL) showed evidence of IS by Q_IgG,_ but not IgG index. All of these also had an elevated kappa index. None of the patients in the Kappa-OCB-group had an IgG IF and/or IgG index indicating IS.

Table 4 shows the demographic and laboratory findings of patients classified on the basis of their diagnostic category. Diagnostic categories were denominated as follows: (1) MS; (2) INFL: other inflammatory but not infectious CNS or PNS disorders; (3) INFECT: infectious CNS or PNS disorders; and (4) Other: miscellaneous, non-inflammatory disorders of CNS/PNS. See Appendix A for diseases/disorders included in the INFL and INECT group.

The highest mean value of the kappa index was observed in MS patients who were significantly younger and more frequently female than the other groups. The kappa index was significantly higher in INFECT patients versus patients with non-inflammatory neurological disorders (Other). INFECT patients had a higher CSF cell count and higher CSF proteins. The mean IgM and IgA index value was significantly higher in INFECT patients than in other groups, and nearly all patients (18/19) presented an elevated IgM index. On the other hand, no difference was observed as regards the mean IgG index value.

### 3.3. IgM and IgA Indexes in Infectious Disorders

Since nearly all patients with INFECT (18/19, 95%) had an elevated IgM index, and more than half of them (10/19) had an elevated IgA index, in a sub-analysis, we calculated the diagnostic performance of the IgM and IgA index in INFECT versus all other diagnoses. The IgM index at the chosen cut-off (0.10) had an AUC of 0.77, with a high sensitivity and negative predictive value (NPV) (95% and 98%, respectively) but a low specificity and positive predictive value (PPV) of 60% and 31%, respectively. However, when applying the Youden index, the best IgM index cut-off turned out to be 0.15 with an AUC of 0.84 (Table 5). The IgA index at the chosen cut-off (0.40) had an AUC of 0.73 with low sensitivity (53%) and high specificity (93%), while the best IgA index cut-off, defined by the Youden index was 0.32 with a sensitivity of 90%, a specificity of 64%, and an AUC of 0.77 (Table 5).

## 4. Discussion

The main findings of the study are as follows.

In patients with an elevated kappa index and without OCBs, the most frequent diagnoses are MS (27/69, 39%), followed by infectious CNS and PNS diseases (16/69, 23%).Approximately two-thirds of the Kappa+OCB- patients present additional evidence of IS, as demonstrated by an elevated IgG/IgM/IgA index or by the presence of IgM OCBs.Patients in the INFECT group had the highest mean IgM and IgA indexes compared with the other groups.

### 4.1. Multiple Sclerosis

In the total cohort, kappa index values of ≥6.10 showed a high sensitivity (93%—43 out of 46 patients) in patients with MS, slightly higher than the one found in a recent systematic review and meta-analysis (weighted average of 88%) [1]. As expected, MS patients were frequent in the Kappa+OCB+ group (18/26, 69% of all Kappa+OCB+ patients). In the latter, IgG IF > 0% had better sensitivity (65%) than the IgG index (58%) in detecting IgG IS.

As said, among Kappa+OCB- patients, MS is the most frequent diagnosis. This is not surprising, since it is known that approximately 40% of patients with clinically definite MS and approximately 20% of clinically isolated syndrome (CIS) patients show intrathecal IgM synthesis, as demonstrated by either IgM OCBs or by the IgM index/IF [13,14].

According to our study, 28% of MS patients (13/46) presented intrathecal IgM synthesis, as shown by an IgM index of ≥0.10 (13/13) or by the presence of IgM OCBs (4/13). This proportion is very similar to the one found in a recent meta-analysis (29% overall) [15]. In MS patients, though, due to its low sensitivity, at disease onset, the IS of IgM is considered useful for prognostic but not for diagnostic purposes.

However, studies have repeatedly shown an association with a worse disease course, for instance with worse radiological outcomes such as a higher number of new/enlarging T2 lesions, T2 lesion counts, and serum neurofilament light chain levels [13,14,16]. Furthermore, various studies have reported a higher T1 lesion load in MS patients with IS of IgM, demonstrated by an elevated IgM index [17,18]. From the clinical point of view, it has been shown that the risk of suffering a second clinical attack was higher in patients with IgM IS [8,17,19]. Furthermore, the role of IgM IS in predicting disability progression, was described in various studies. Patients with IgM OCBs present an increased risk of reaching an EDSS of 4 [20] and significantly higher EDSS scores after 5 [13] and 7 years [18] of follow-up.

IgA IS was present in a small proportion of MS patients (7%). A similar result (14%) was observed in a large German study [21]. A slightly higher proportion (24%) of primary progressive MS patients was shown to have IgA IS defined as an IgA index of >0.34 [22]. Contrarily, a recent Spanish study, using a highly sensitive assay based on IEF, found IgA IS in as many as 43% of MS patients [23].

### 4.2. Infectious CNS Disorders

Several studies have underlined that the kappa index may be elevated in patients with INFECT, especially in patients with neuroborreliosis [24,25], but also in other infectious CNS diseases such as HIV, tick-borne encephalitis, and neurosyphilis [2,26,27,28], often having a higher sensitivity than OCBs. In these patients, IgM intrathecal synthesis contributes to an elevated kappa index value. This may be explained by the fact that the primary immune response to an infectious agent is mediated by IgM [29]. The linear IgM index showed good sensitivity and NPV for INFECT versus the remaining diagnoses in this present study.

IgA IS is also involved in the first line of defence against various viruses such as SARS-CoV-2, influenza virus, or HIV [30,31,32], and our study, which included various infectious disorders of CNS/PNS, found that one-third of patients had a positive IgA index.

Combining theIgM/IgA indexes with IgM OCBs yielded a very high sensitivity in diagnosing INFECT (95%) and is consistent with the elevation of the kappa index in this group of patients and with the underlying pathogenesis of this type of CNS disorder in its early stages, which is largely IgM- and IgA-dependent.

### 4.3. Autoimmune Encephalitis and Other Inflammatory CNS Disorders

Among INFL, the most frequent disorder was autoimmune encephalitis (AE) (eight cases, 7% of all included patients and 31% of INFL). In this AE cohort, the mean kappa index was 6.51 ± 5.39, with 3/8 (38%) patients presenting a kappa index of ≥6.00 and 6/8 (75%) patients presenting a kappa index of ≥3.00. This cohort is in partial overlap (n = 5 patients) with a cohort described in a recent study by our group on 34 patients with AE, which showed that a kappa index of ≥6.00 was present in 29% of patients and that the best kappa index cut-off to discriminate AE from non-inflammatory neurological diseases was 3.00. A kappa index of ≥3.00 was present in 50% of patients with definite AE, whereas only 34% of these had OCBs. We therefore suggested that the kappa index could be useful as a more sensitive marker of IS and as a supportive marker of neuroinflammation in the diagnostic work-up of suspected AE [33].

On this topic, a large German study (n = 346) showed that 32/84 patients (38%) with non-MS INFL/INFECT had a kappa index of >5.90, The sensitivity of this cut-off in patients with AE was 13% (1/8 patients) [28]. The same study reported intrathecal IgM synthesis in 25% (2/8) of patients with AE.

As concerns non-MS demyelinating disorders (MOGAD and ADEM), in our cohort, only three patients with this type of diagnosis were in the Kappa+OCB- group of patients. An elevated kappa index was reported to be relatively rare in MOGAD patients, with a frequency of about 15% according to a recent French study [34]. The aforementioned study investigated the presence of OCBs and/or an elevated kappa index in a cohort of 40 patients with MOGAD. The same study showed that OCBs were found less frequently (4/40, 10%) than an elevated kappa index. Moreover, it was reported that the mean value of the kappa index was higher in patients with anti-AQP4 antibodies (16.80 versus 1.30) compared with MOGAD patients.

### 4.4. Limits of the Study

The limits of the study are its retrospective nature and the relatively small size of the control groups, which included randomly selected patients, as opposed to all consecutive patients undergoing a diagnostic spinal tap throughout the two-year study period.

## 5. Conclusions

Kappa+OCB- patients mostly had a diagnosis of MS or of an infectious CNS disease. In this group of patients, the IS of IgG, IgM, or IgA can co-occur. Furthermore, the IS of IgM/IgA was particularly pronounced in patients with infectious CNS disorders, irrespective of the kappa index/OCB status.

## Figures and Tables

**Figure 1 biomolecules-15-00090-f001:**
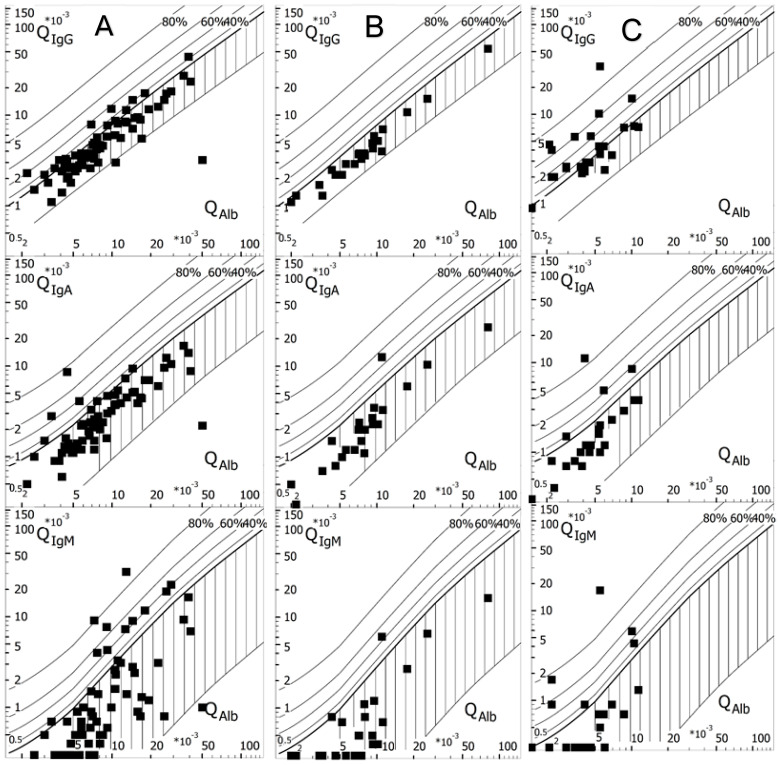
Reibergrams of Kappa+OCB- (**A**), Kappa-OCB- (**B**), and Kappa+OCB+ (**C**) patients. The quotients are indicated with black squares in relation to their CSF/serum albumin quotients (Qalb). IgG/M/A intrathecal synthesis can be assumed when the Q kappa is above the bold line, depicting the hyperbolic border line of Q Ig (lim). The grey percentile lines with 40%, 60%, and 80% represent the fraction of IgG/M/A that has been synthesized in the CSF in relation to the total amount of the molecule in the CSF. Contrarily, values falling into the striped area of the graph are not indicative of IgG/M/A IS. The lower line indicates the lower limit of the reference range (Q Ig low). Modified version of graphs created using the free software available at Albaum (www.albaum.it, accessed on 18 June 2024).

**Table 1 biomolecules-15-00090-t001:** Diagnoses of Kappa+OCB- patients (n = 69).

Diagnosis	Number of Patients	%
MS	27	39
Infectious encephalitis	14	20
Isolated acute transverse myelitis (non-infectious)	5	7
CIDP + GBS	3	4
Meningeal carcinomatosis	2	3
MOGAD	2	3
Cerebral lymphoma	2	3
Neurosarcoidosis	2	3
ADEM	1	1
Other diagnoses *	11	16

MOGAD: myelin oligodendrocyte glycoprotein antibody-associated disease; ADEM: acute disseminated encephalomyelitis; CIDP: chronic inflammatory demyelinating polyneuropathy; GBS: Guillan–Barré syndrome. * One case each of amyotrophic lateral sclerosis, progressive supranuclear palsy, stroke, autoimmune encephalitis, Borrelia polyradicultis, neurinoma, Degos’ disease, migraine with aura, Neuro-Behçet, cerebral venous thrombosis, and infectious myeloradiculitis.

**Table 2 biomolecules-15-00090-t002:** Demographic and laboratory findings in the overall population and across patients grouped on the basis of their kappa index/OCB status.

	Overall Population (n = 119)	Kappa+OCB- (n = 69)	Kappa-OCB- (n = 24)	Kappa+ OCB+ (n = 26)	*p*-Value
Age in years (mean ± SD)	55 ± 19	47 ± 18	60 ± 20	38 ± 18	<0.001
Sex	49% M 51% F	45% M 55% F	58% M 42% F	50% M 50% F	NS
Kappa index (mean ± SD)	35.25 ± 83.66	16.37 ± 19.72	1.33 ± 1.59	116.67 ± 151.64	<0.001
IgG index (mean ± SD)	0.71 ± 0.59	0.62 ± 0.19	0.52 ± 0.81	1.14 ± 1.14	<0.001
Patients with elevated IgG index, n (%)	32 (27)	17 (25)	0 (0)	15 (58)	<0.001
IgM index (mean ± SD)	0.21 ± 0.39	0.22 ± 0.36	0.11 ± 0.11	0.27 ± 0.60	NS
Patients with elevated IgM index, n (%)	58 (49)	36 (52)	10 (42)	12 (46)	NS
IgM OCB-positive patients, n (%)	29 (25)	19 (28) (n = 68)	4 (17)	6 (23)	NS
IgA index (mean ± SD)	0.57 ± 2.41	0.73 ± 3.14	0.28 ± 0.21 (n = 23)	0.38 ± 0.51	<0.017
Patients with elevated IgA index, n (%)	17 (14)	11 (16)	1 (4) (n = 23)	5 (19)	NS
Percentage of patients with positive IgM/IgA index or IgM OCBs	50	52	42	50	NS
Percentage of patients with positive IgG/IgM/IgA index or IgM OCBs	60	59	42	77	0.040

SD: standard deviation; F: female; M: male; NS: not significant. The results of the post hoc analysis can be found in the Appendix A.

**Table 3 biomolecules-15-00090-t003:** Patients with IF > 0, according to Reiber.

	Overall Population (n = 119)	Kappa+OCB- (n = 69)	Kappa-OCB- (n = 24)	Kappa+ OCB+ (n = 26)	*p*-Value
IgG IF > 0 according to Reiber, n (%)	31 (26)	14 (20)	0 (0)	17 (65)	<0.001 *
IgM IF > 0 according to Reiber, n (%)	18 (15)	11 (16)	1 (4)	6 (23)	NS
IgA IF > 0 according to Reiber, n (%)	12 (10)	7 (10)	1 (4) (n = 23)	4 (15)	NS

* Statistically significant differences in post hoc tests: Kappa+OCB+ versus Kappa+OCB- (*p* < 0.001); Kappa+OCB+ versus Kappa-OCB- (*p* < 0.001); Kappa+OCB- versus Kappa-OCB- (*p* = 0.017); IF: intrathecal fraction; NS: not significant.

**Table 4 biomolecules-15-00090-t004:** Demographic and laboratory findings in patients according to their diagnostic category.

	MS (n = 46)	INFL (n = 26)	INFECT (n = 19)	Other (n = 28)	*p*-Value
Age in years (mean ± SD)	38 ± 15	51 ± 17	49 ± 24	59 ± 19	<0.001
Sex	27% M 73% F	50% M 40% F	63% M 37% F	71% M 29% F	0.002
Kappa index (mean ± SD)	64.48 ± 120.07	12.83 ± 24.47	36.84 ± 68.57	6.96 ± 10.06	<0.001
Patients with kappa index ≥ 5.0, n (%)	45 (95)	19 (73)	19 (100)	12 (43)	<0.001
IgG index (mean ± SD)	0.72 ± 0.41	0.59 ± 0.11	1.02 ± 1.28	0.61 ± 0.22	NS
Patients with elevated IgG index, n (%)	17 (37)	4 (15)	8 (42)	3 (11)	0.016
IgM index (mean ± SD)	0.10 ± 0.11	0.12 ± 0.11	0.65 ± 0.79	0.17 ± 0.25	<0.001
Patients with elevated IgM index, n (%)	13 (28)	12 (46)	18 (95)	15 (54)	<0.001
IgA index (mean ± SD)	0.37 ± 0.45	0.33 ± 0.19 (n = 25)	1.74 ± 5.96	0.30 ± 0.17	<0.006
Patients with elevated IgA index, n (%)	3 (7)	1 (4) (n = 25)	10 (53)	3 (11)	<0.001
IgM OCB-positive patients, n (%)	4 (9)	5 (19)	11 (58)	9 (33) (n = 27)	<0.001
Patients with positive IgM/IgA index or IgM OCBs, n (%)	13 (28)	13 (50)	18 (95)	16 (57)	<0.001
Patients with positive IgG/IgM/IgA index or IgM OCBs, n (%)	23 (50)	15 (58)	18 (95)	16 (57)	0.009
CSF cell count, median (interquartile range) (cells/µL)	3 (1–4)	5 (2–25)	20 (7–90)	2 (1–13)	<0.001
CSF proteins, median (interquartile range) (cells/µL)	38 (30–46)	56 (42–65)	84 (56–100)	65 (40–91)	<0.001
QAlb × 100 (mean ± SD)	0.58 ± 0.40	0.89 ± 0.66	1.21 ± 0.73	1.38 ± 1.63	<0.001

SD: standard deviation; F: female; M: male; NS: not significant. The results of the post hoc analysis can be found in the Appendix A.

**Table 5 biomolecules-15-00090-t005:** Diagnostic performance of the kappa index and of the IgM and IgA indexes for both the initially chosen cut-offs (0.10 and 0.40, respectively) and for the best cut-off according to the Youden index (0.15 and 0.32, respectively) in INFECT patients versus all other diagnoses.

	Sensitivity (95% CI)	Specificity (95% CI)	PPV (95% CI)	NPV (95% CI)	AUC (95% CI)
IgM index ≥ 0.10	95 (85–100)	60 (50–70)	31 (19–43)	98 (95–100)	0.77 (0.70–0.84)
IgM index ≥ 0.15	89 (76–100)	79 (71–87)	45 (29–61)	98 (94–100)	0.84 (0.76–0.92)
IgA index ≥ 0.40	53 (30–75)	93 (88–98)	59 (35–82)	91 (86–97)	0.73 (0.59–0.87)
IgA index ≥ 0.32	90 (76–100)	64 (54–73)	32 (20–45)	97 (93–100)	0.77 (0.67–0.84)
Kappa index ≥ 5.0	100 (100–100)	24 (24–24)	20 (20–20)	100 (100–100)	0.62 (0.58–0.66)

PPV: positive predictive value, NPV: negative predictive value, AUC: area under the receiver operating characteristic curve; CI: confidence interval.

## Data Availability

The data underlying the analyses presented will be shared upon reason request.

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
