# Peer review of "Elevated Kappa Index in the Absence of Cerebrospinal Fluid IgG Oligoclonal Bands: Contribution of Intrathecal IgM and IgA Synthesis"

_biomolecules, 2025, doi:10.3390/biom15010090_

Round 1

Reviewer 1 Report

Comments and Suggestions for Authors

The aim of the study was to give answers to questions that both neurologic clinics and CSF laboratories have to face quite often, that is “what is the clinical significance of an elevated kappa index in a negative OCB patient, are there additional tests that could be proved useful in these cases, are there specific diagnoses correlated with this finding?”. For that reason, the authors provide an adequate amount of data from neurological patients that had underwent lumbar puncture and CSF analysis for the detection of qualitative and quantitative intrathecal immunoglobulin synthesis. At the end, the authors seem to succeed in giving these answers, however there are several contradictory data that have to be clarified and, in some cases, additional information is needed.

Specific comments

Line 94: Which was the modification that has been made on Villar’s methodology?

Line 99: Was there a Kappa-OCB+ group? Please, make a comment.

Line 116, Table 1:

·      The title “Diagnosis” of the first column is incorrect: parkinsonism, migraine, headache are not diagnoses. Please, reconsider.

·      What was the specific pathogens for the 3 cases of infectious encephalitis? Why the neuroborreliosis case was not included, along with infectious myeloradiculitis, in a single group under the name “CNS infections”? A classification of “diagnoses” as “autoimmune disorders”, neurodegenerative disorders”, “CNS diseases of vascular etiology” etc. would be more comprehensive.

Line 143, figure 1: This is a nice presentation of the FLC Reibergrams. However, how were they performed? Did they perform better than the Ig indices as expected? How you comment on that? Are both methods for the quantitative measurement of intrathecal immunoglobulin synthesis necessary? Why there is no comment about Reibergrams in the Discussion section and they are not mentioned in the Introduction section?

Line 157, table 4:

·      Please be specific about the diagnoses included under the titles of the columns (INFL, INFECT etc.) which should be in agreement with table 1.

·      There is a contradiction in the category OTHER with a mean value for CSF cell count 34.91 while inflammatory disorders are not a part of it. Please, explain.

Comments on the Quality of English Language

Quality of English Language should be improved.

Author Response

 1) Line 94: Which was the modification that has been made on Villar’s methodology?

Response

Dear reviewer, thank you for permitting us to clarify. Compared to Villar’s methodology the main modifications are: agarose isoelectric focusing (IEF) gel is prepared with 1.5 ml of Pharmalyte pH 5-8 and 2.0 ml of Pharmalyte pH 3-10; 10 µl paired samples are applied using a sodium dodecyl sulphate ( SDS) applicator strips; polyvinylidene fluoride (PVDF) membrane is wetted in ethanol and then washed in 2 changes saline solution; the membrane is incubated with rabbit anti human IgM and then with alkaline phosphatase (AP)-conjugated goat anti-rabbit. This information has been added to the supplementary material (S1).

2) Line 99: Was there a Kappa-OCB+ group? Please, make a comment.

Response

Dear reviewer, Kappa-OCB+ patients were not included in this study. The concordance between OCBs and kappa index is around 90%. Discrepant cases are mostly patients with a positive kappa index and negative OCBs. For example, in a previous study (Ferraro et al. Diagnostics, 2020), OCBs and kappa index were discrepant only in 56 out of 540 cases. The majority (41/56, 73%) had a positive kappa index and absent OCBs, and only 15 had OCBs and a kappa index<5.8, amounting to 2.8% of the total cohort. We did not include a control group with these characteristics since the numerosity of this group would have been quite low due to the rarity of this pattern, which, additionally, does not generally cause difficulties in interpretation.

3) Line 116, Table 1: The title “Diagnosis” of the first column is incorrect: parkinsonism, migraine, headache are not diagnoses. Please, reconsider.

Response

We thank the reviewer for this comment since it prompted and updated revision of patients’ clinical charts. This has permitted us to specify the correct diagnoses (e.g. the patient with “migraine” had a diagnosis of migraine with aura, the patient with “headache” had a diagnosis of tension-type headache, one patient with “parkinsonism” had Progressive Supranuclear Palsy). Furthermore, it permitted us to ascertain that a cerebral lymphoma was diagnosed in one patient presenting with “parkinsonism”, and that an underlying neurosyphilis was detected in a patient with status epilepticus. Consequently, all tables have been updated and one patient was moved from OTHER to INFECT (status epilepticus due to neurosyphilis).

4) What was the specific pathogens for the 3 cases of infectious encephalitis?

Response

Dear reviewer, the infectious agent was unidentified in three cases of infectious encephalitis. This has been added in the legend to S2. This diagnosis was, however, deemed probable based on clinical grounds and on compatible CSF data (pleiocytosis, elevated protein count).

5) Why the neuroborreliosis case was not included, along with infectious myeloradiculitis, in a single group under the name “CNS infections”? A classification of “diagnoses” as “autoimmune disorders”, neurodegenerative disorders”, “CNS diseases of vascular etiology” etc. would be more comprehensive.

Response

When comparing different groups of patients (INFL versus INFECT versus OTHER), we grouped different diagnoses together. For example, as suggested, “INFECT” comprises infections of both the CNS and the PNS, autoimmune disorders are grouped under “INFL”, together with inflammatory CNS and PNS disorders, and neurodegenerative disorders and diseases of vascular etiology are grouped under “OTHER”. Please note that, to this end, 3 patients were moved from OTHER to INFL, which now also comprises inflammatory PNS disorders (Guillan-Barré syndrome and Chronic Inflammatory Demyelinating Polyneuropathy).

However, tables illustrating the different diagnoses/disorders (Table 1 for the kappa+/OCB- group and S2 for the overall population), list them based on their frequency, to give the reader a more detailed overview of the cohort characteristics. For example, information on the number and frequency of autoimmune encephalitides or MOGAD would have been lost if grouped together as “autoimmune disorders”. And this type of information may be of interest to some readers. We have, in any case, added information on which disorders were grouped together in INFL and INFECT in the supplementary material (S2).

6) Line 143, figure 1: This is a nice presentation of the FLC Reibergrams. However, how were they performed?

Response

Dear reviewer, Reibergrams were not applied to kappa FLC but to IgG, IgM and IgA (since the scope of the study was to ascertain the contribution if IgM/IgA intrathecal synthesis in the kappa+/OCB- patients), using the known Reiber formulas and the freeware software available at www.albaum.it to generate the figures. Information on the performance of the FLC Reibergram (slightly more sensitive but greatly less specific than the kappa index can be found in our previous study (Ferraro et al, Diagnostics; 2020).

7) Did they perform better than the Ig indices as expected? How you comment on that? Are both methods for the quantitative measurement of intrathecal immunoglobulin synthesis necessary? Why there is no comment about Reibergrams in the Discussion section, and they are not mentioned in the Introduction section?

Response

We are not sure if these comments were thought for FLC Reibergrams (which were not carried out) or for the IgG/IgA/IgM Reibergrams. The following responses pertain the IgG/IgA/IgM Reibergrams included in the study.

With regard to the performance of the Reiber formulas for IgG/IgA/IgM: as expected, in the OCB+ group, the Reiber formula for IgG was more sensitive (65%) than the linear IgG index (58%) in detecting intrathecal IgG synthesis. Furthermore, consistent with literature data (e.g. Monreal et al, Neurol  Neuroimmunol Neuroinflamm; 2021), using the cut-off of 0.1 for the IgM index, the Reiber formula for IgM  was less sensitive (23%) than the linear IgM index (46%), while the performance was similar for the IgA index and IgA Reiber formula. In the kappa+/OCB- group there were no relevant differences between the two methods for IgG and IgA, while a higher sensitivity of the IgM index versus the Reiber formula was confirmed for IgM.

We have added a brief comment on the performance of the IgG index/Reiber formula in the discussion section, but did not discuss these aspects (differences between quantitative measures of IgG/A/M) in depth as they were beyond the scope of the study and since we suspect that the reviewer’s comments were addressed to FLC Reibergrams..

Of note is that, in the overall population, all discordant cases between IgG index and the Reiber formula, had a positive kappa index. This has been added to the results section.

We would like to explain that the main reason we added the IgG/IgA/IgM Reibergrams to the manuscript was to depict, in a visual, rapidly comprehensive manner, how in the kappa+/OCB- group, contrarily to the other groups, there is a much larger proportion of patients with an IF>0% for IgM and IgA, confirming an intrathecal synthesis of Ig classes other than IgG in this group. However, if not considered useful, the information and figures relating to the Reiber formulas and Reibergrams can be eliminated.

Finally, the introduction does, actually mention that we “assessed the intrathecal synthesis of IgM and IgA in these patients using linear and hyperbolic indexes of IS”.

8) Line 157, table 4:Please be specific about the diagnoses included under the titles of the columns (INFL, INFECT etc.) which should be in agreement with table 1.

Response

Dear reviewer, thank you for this comment. Additional information concerning single diagnoses of each group has been added to the supplementary material (S2).

9) There is a contradiction in the category OTHER with a mean value for CSF cell count 34.91 while inflammatory disorders are not a part of it. Please, explain.

Response

Dear reviewer, thank you for pointing out this unexpected finding which is due to some outliers. Given the presence of some outliers (with diagnoses of cerebral venous sinus thrombosis, cerebral lymphoma, leptomeningeal carcinomatosis) and low total number of patients, we have decided to express the CSF cell count and protein count as medians with interquartile ranges as such presentation of the data is more appropriate and better illustrates the examined population.

Reviewer 2 Report

Comments and Suggestions for Authors

The authors aimed to characterize a group of patients with elevated CSF kappa index without (IgG) oligoclonal bands (OCB). The study draws attention to the potential clinical utility of kappa index, which may reflect intrathecal immunoglobulin synthesis. However, some issues regarding this manuscript need to be addressed or clarified.    1. It is notable that the study did not include patients with positive OCB and normal kappa index. The reason for it should be provided. Subjects with both positive OCB and elevated kappa index were "randomly selected" for this study. My opinion is that if patients were consecutively recruited, all those with either positive OCB or elevated kappa index (or both) should be enrolled and their data presented, so that readers could see the big picture regarding the CSF profiles in the study population.    2. The diagnostic entities in Table 1 are not mutually exclusive. For example, both ADEM and isolated acute transverse myelitis could also be MOGAD. And how about the anatomical details of the two cases diagnosed with MOGAD?    3. Table 1, what was the indication for CSF study in the patient with Parkinsonism?   4. Some data in Table 2 are apparently incorrect (such as some percentage data). Please revise the Table thoroughly.    5. Table 4, it is surprising that the highest mean IgG index and second highest CSF protein occurred in "OTHER" group, which presumably consisted of patients with non-inflammatory nervous system disorders. Could the authors provide an explanation for these findings?   6. Table 4, I would suggest that data regarding patients with elevated kappa index are also compared across different conditions. I suspect that positive IgM or IgA index or IgM OCB might be more sensitive and specific for CNS infection as compared to elevated kappa index.    7. It would be better if supplementary material 2 and 3 are presented in tabular forms.    8. Some references, such as ref 16 and ref 32 and ref 33, are not correctly presented. 

Comments on the Quality of English Language

Some abbreviation, like IS, is not commonly used and may not be needed.

Author Response

  1. It is notable that the study did not include patients with positive OCB and normal kappa index. The reason for it should be provided. Subjects with both positive OCB and elevated kappa index were "randomly selected" for this study. My opinion is that if patients were consecutively recruited, all those with either positive OCB or elevated kappa index (or both) should be enrolled and their data presented, so that readers could see the big picture regarding the CSF profiles in the study population.   

Response

Dear reviewer, thank you for this comment. With regard to the inclusion of patients with OCB and negative kappa index, as also stated in response to reviewer nr. 1: the concordance between OCBs and kappa index is around 90%. Discrepant cases are mostly patients with a positive kappa index and negative OCBs. For example, in a previous study (Ferraro et al. Diagnostics, 2020), OCBs and kappa index were discrepant only in 56 out of 540 cases. The majority (41/56, 73%) had a positive kappa index and absent OCBs, and only 15 had OCBs and a kappa index<5.8, amounting to 2.8% of the total cohort. We did not include a control group with these characteristics since the numerosity of this group would have been quite low due to the rarity of this pattern, which, additionally, does not generally cause difficulties in interpretation.

We randomly selected patients with different OCB/kappa index profiles solely for the purpose of having control groups for the kappa+/OCB- group, which was the focus of the present study. The bigger picture was provided in a study by our group which did, indeed, characterize OCB and kappa index in all consecutive patients (nr=540) undergoing a spinal tap for any diagnostic reason throughout a two-year period (Ferraro et al. Diagnostics, 2020).

  1. The diagnostic entities in Table 1 are not mutually exclusive. For example, both ADEM and isolated acute transverse myelitis could also be MOGAD. And how about the anatomical details of the two cases diagnosed with MOGAD?   

Response

Dear reviewer, patients diagnosed with MOGAD (one with presence of spinal and brainstem lesions compatible with MOGAD and presence of serum anti-MOG antibodies and the other with longitudinally extensive transverse myelitis and bilateral evidence of optic nerve inflammation together with presence of serum anti-MOG antibodies) fulfilled the 2023 diagnostic criteria for MOGAD (Banwell et al. 2023). The cases of ADEM and isolated acute transverse myelitis could not be diagnosed as MOGAD due to the absence of anti-MOG antibodies.

  1. Table 1, what was the indication for CSF study in the patient with Parkinsonism?  

Response

Dear reviewer, CSF study was performed in this case due to concurring cognitive complaints in order to assess the dosage of tau, phosphorylated tau and amyloid B-42.

  1. Some data in Table 2 are apparently incorrect (such as some percentage data). Please revise the Table thoroughly.   

Response

Dear reviewer, thank you for pointing out some inconsistencies in the table. We have thoroughly revised the table and specified that for one patient the data on IgAs was missing, while for another the IgM IEF was missing.

  1. Table 4, it is surprising that the highest mean IgG index and second highest CSF protein occurred in "OTHER" group, which presumably consisted of patients with non-inflammatory nervous system disorders. Could the authors provide an explanation for these findings?  

Response

Dear reviewer, thank you for this important observation. After an updated patient clinical chart revision (see also response to Reviewer nr.1), following both your comments, we were able to ascertain that neurosyphilis was diagnosed in a patient with status epilepticus. This patient, who was initially included in the “OTHER” group,  had a very high IgG index (6.22) and protein count (93mg/dl). Furthermore, three patients with PNS disorders (Guillan-Barré syndrome and Chronic Inflammatory Demyelinating Polyneuropathy) who typically have an elevated CSF protein count, have now been appropriately moved to the “INFL” group.  Following these corrections, IgG index is no longer unexpectedly high. As regards the CSF protein count in the “OTHER” group, this could be explained by the mean higher age, by the inclusion of disorders characterized by blood-brain-barrier damage (acute stroke, status epilepticus), and by the presence of some outliers (with diagnoses of leptomeningeal carcinomatosis, cerebral lymphoma). Considering this aspect, in addition to the low number of patients, we have decided to express the CSF cell count and protein count as medians with interquartile ranges, to better depict the characteristics of the examined population.

  1. Table 4, I would suggest that data regarding patients with elevated kappa index are also compared across different conditions. I suspect that positive IgM or IgA index or IgM OCB might be more sensitive and specific for CNS infection as compared to elevated kappa index.   

Response

Dear reviewer, thank you for this observation. Indeed, we sought to determine the diagnostic accuracy of IgM/IgA IS in predicting INFECT (the vast majority of which,16/19, belonging to the kappa+/OCB- group) among all patients. Kappa index was positive in 100% of patients with INFECT while elevated IgM/IgA indices or IgM OCB were present in 95%. Following your suggestion, we calculated the diagnostic accuracy of the kappa index in predicting INFECT (see below). If considered useful this can be added to the results section. As can be seen, kappa index had a sensitivity of 100% but, as expected (considering the presence of MS and INFL), a very low specificity compared to markers of IgM/IgA IS.

Diagnostic performance of kappa index in INFECT versus all other diagnoses.

Sensitivity

(95% CI)

Specificity

(95% CI)

PPV

(95% CI)

NPV

(95% CI)

AUC

(95% CI)

Kappa index ≥5.0

100 (100-100)

24 (24-24)

20 (20-20)

100 (100-100)

0.62 (0.58-0.66)

PPV: positive predictive value, NPV: negative predictive value, AUC: area under the receiver operating characteristic curve; CI: confidence interval

  1. It would be better if supplementary material 2 and 3 are presented in tabular forms. 

Response

The supplementary material has been transformed into tabular forms.

  1. Some references, such as ref 16 and ref 32 and ref 33, are not correctly presented. 

Response

The references have been corrected.

Reviewer 3 Report

Comments and Suggestions for Authors

This is a well written article and the topic is of clinical interest.  The main goal of the study was to characterize patient phenotypes associated with elevated Kappa Free Light Chains but negative OCBs.  Although the most prevalent diagnosis in such a cohort of patients was still MS, as expected, a high percentage of subjects showed positivity for IgM and IgA.  Using ROC methods, certain IgM and IgA cut-off values showed high predictability for CNS infections.  These findings are clinically useful.  

Author Response

We thank the reviewer for this comment.

Round 2

Reviewer 1 Report

Comments and Suggestions for Authors

Congratulations for this interesting and very useful in clinical practice study.

Author Response

We thank you for this comment.

Reviewer 2 Report

Comments and Suggestions for Authors 1. The authors respond to my previous comment (first point) "We randomly selected patients with different OCB/kappa index profiles solely for the purpose of having control groups for the kappa+/OCB- group, which was the focus of the present study." Actually, my opinion is that "selection" itself, as opposed to "inclusion of all cases", may introduce bias in the assessment of diagnostic performance. For example, even the "Kappa-OCB-" controls were randomly selected, the number of cases selected will affect the results of some diagnostic performance indicators.   2. It would be better to include the distribution of diagnoses in Kappa+OCB+ and Kappa-OCB- groups also in Table 1.   3. It would be better to include kappa index in Table 5 for better comparisons across different indicators in this context.

Author Response

  1. The authors respond to my previous comment (first point) "We randomly selected patients with different OCB/kappa index profiles solely for the purpose of having control groups for the kappa+/OCB- group, which was the focus of the present study." Actually, my opinion is that "selection" itself, as opposed to "inclusion of all cases", may introduce bias in the assessment of diagnostic performance. For example, even the "Kappa-OCB-" controls were randomly selected, the number of cases selected will affect the results of some diagnostic performance indicators.  

Response

Dear reviewer, thank you for permitting us to further clarify this aspect.

Our previous study (Ferraro et al, Diagnostics; 2020) enrolled all consecutive patients carrying out a spinal tap for any reason throughout a two-year period. A total of 540 patients were enrolled. The tables below show the distribution of the different diagnostic categories in the Kappa+/OCB+ and in the Kappa-/OCB- groups of the above-mentioned study compared to those of the present study. The overall chi-square test is non significant in both cases (p=0.33 for proportions among Kappa-/OCB- patients and p=0.36 for Kappa+/OCB+ patients). Post-hoc pairwise comparisons using the Fisher’s exact test are shown in the tables,

Since our study has been using the kappa index in routine clinical practice since 2018, including all cases would have entailed including well over a thousand patients as controls, which would not have been feasible due to the costs of assessing the IgM/IgA intrathecal synthesis (via IgM/IgA index and IgM OCB), which is not in current routine clinical practice. Since the necessary Freelite kits were provided by Binding Site, we are not able, at the moment, to implement further analyses. We hope that the control groups can be considered adequate thanks to this demonstration.

Furthermore, we would like to draw the attention to the main aim of the study, which was assessing the diagnoses associated with a Kappa+/OCB- profile and the presence of IgM/IgA intrathecal synthesis in this group. Diagnostic performance assessments (paragraph 3.3, which can be eliminated if deemed necessary) were carried out between different diagnostic categories, irrespective of their Kappa/OCB status, solely as a sub-analysis of a secondary outcome.

1) Diagnostic categories among Kappa-/OCB- patients

Ferraro et al, 2020

(n=396 )

Present study

(n=24)

p-value

MS

10 (3%)

1 (4%)

0.481

INFL

59 (15%)

7 (29%)

0.079

INFECT

24 (6%)

0 (0%)

0.384

OTHER

303 (77%)

16 (67%)

0.324

2) Diagnostic categories among Kappa+/OCB+ patients

Ferraro et al, 2020

(n=95)

Present study

(n=26)

p-value

MS

71 (75%)

18 (69%)

0.619

INFL

5 (5%)

4 (15%)

0.098

INFECT

8 (8%)

2 (8%)

1.000

OTHER

11 (12%)

2 (8%)

0.732

  1. It would be better to include the distribution of diagnoses in Kappa+OCB+ and Kappa-OCB- groups also in Table 1.  

Response

Dear reviewer, Table 1 shows the single most frequent diagnoses in the group of interest (with a Kappa+/OCB- profile). As a consequence, adding data on the frequency of these single diagnoses in the other two groups would have yielded a lot of cells containing 0 or 1 patient (patients cluster in “MS” in the Kappa+/OCB+ group and in “other” in the Kappa-/OCB- group). We have, therefore, added the requested information in S2, which now regards the distribution of the most frequent diagnoses in the overall population and in the Kappa+/OCB+ and Kappa-/OCB- groups.

  1. It would be better to include kappa index in Table 5 for better comparisons across different indicators in this context.

Response

Kappa index has been included in Table 5.